# Predict the Gelling Properties of Alkali-Induced Egg White Gel Based on the Freshness of Duck Eggs

**DOI:** 10.3390/foods12214028

**Published:** 2023-11-04

**Authors:** Jun Sun, Jialei Wang, Wan Lin, Baochang Li, Ruipeng Ma, Yuqian Huang, Mohammed Obadi

**Affiliations:** School of Food and Biological Engineering, Jiangsu University, 301 Xuefu Road, Zhenjiang 212013, China; wangjialei199811@163.com (J.W.); 15378267932@163.com (W.L.); libaochang0314@163.com (B.L.); 18632213192@163.com (R.M.); 18855107885@163.com (Y.H.); obadialariki@gmail.com (M.O.)

**Keywords:** egg, gelling properties, alkali-induced egg white gel, comprehensive freshness index, principal component analysis

## Abstract

Preserved egg white (PEW) has excellent gelling properties but is susceptible to the freshness of raw eggs. In this study, the correlation between the comprehensive freshness index (CFI) of raw eggs and the gelling properties of alkali-induced egg white gel (EWG) was elucidated. Results showed that the CFI, established by a principal component analysis (PCA) and stepwise regression analysis (SRA) methods, can be used to predict the freshness of duck eggs under storage conditions of 25 °C and 4 °C. A correlation analysis demonstrated that the CFI showed a strong negative correlation with the hardness and chewiness of alkali-induced EWG and a strong positive correlation with resilience within 12 days of storage at 25 °C and 20 days at 4 °C (*p* < 0.01). It might be due to the decrease in α-helix and disulfide bonds, as well as the hydrophobic interactions showing a first decrease and then an increase within the tested days. This study can provide an important theoretical basis for preserved egg pickling.

## 1. Introduction

As a unique traditional egg product in China, preserved eggs have the concomitant functions of both medicine and food [1,2,3], making them popular with the public. Preserved egg white (PEW) has excellent gelling properties, and the formation of PEW is mainly due to the denaturation, unfolding, and aggregation of egg white protein (EWP) induced by alkali, which ultimately forms a three-dimensional gel network structure [4]. However, during egg storage, the freshness of the eggs gradually decreases with time [5]. On the one hand, the water and CO_2_ inside the eggs will gradually transfer to the outside environment through the pores of the eggshells, which directly leads to the rise of albumen pH and the expansion of the air chamber [6]. At the same time, the rise in albumen pH would promote protein hydration and hydrolysis of the concentrated protein [7], thus changing the aggregation behavior of the proteins in the eggs. On the other hand, the microorganisms outside the eggshell will gradually transfer to the inside of the eggs through the eggshell’s pores, and the microorganisms will break down the albumin proteins, which can also encourage the thinning of the concentrated proteins [8].

In addition, the molecular structure of EWP will also undergo continuous changes during storage. According to research on the molecular structure of EWP during storage [9], the content of the random coil structure of egg white protein after storage for 0 days accounted for 20.14%, while the content of the random coil structure increased to 24.02% after storage for 50 days, the α-helix and β-sheet decreased from 29.69% and 29.84% to 25.94% and 27.87%, respectively, and the free sulfhydryl group (SH) increased from 1.02 µmol/g to 1.57 µmol/g, while the disulfide bond (S–S bond) decreased from 28.85 µmol/g to 24.16 µmol/g. In short, eggs with different storage days have different microbial quality, aggregation behaviors and spatial conformations.

Research found that different initial spatial structure, denaturation, and aggregation behaviors of EWP can induce it to form different types and textural properties of alkali-induced EWG, and that’s why the freshness of raw eggs will significantly affect the gelling properties of PEW [10]. However, the effects of the freshness of duck eggs on the gelling properties of PEW still need further study. First, at present, in the research on the effects of the freshness of raw eggs on the gelling properties of PEW, the microbial quality and damage rate of the products were not considered, so the raw duck eggs were stored for 40 days at room temperature [11]. In addition, there are many indicators for evaluating the freshness of raw eggs, such as Haugh unit, yolk index, albumen pH, chamber height, and so on [12,13], and the correlation between the freshness of raw eggs and the gelling properties of alkali-induced EWG has not been established.

Currently, due to the large number of freshness evaluation indicators for raw eggs, researchers can combine numerous freshness evaluation indicators into a CFI to evaluate the freshness of the raw egg. For example, DU and Yimenu [14,15] selected HU, eggshell thickness, shell breaking strength and deformation, albumen height, and yolk film strength as research indicators to establish a prediction model for eggs during storage, which provided a theoretical basis for egg freshness detection. Although polynomial models have been developed for egg freshness, the relationship between the predictive model of raw egg freshness and the quality of alkaline-induced egg white gel was not investigated.

Thus, in this study, the methods of principal component analysis (PCA) and stepwise regression analysis (SRA) were used to reduce the dimensionality of the freshness data and establish a CFI for raw eggs. On this basis, the correlation between the CFI of duck eggs and the gelling properties of alkali-induced EWG was elucidated to provide an important theoretical basis for the selection of raw eggs for preserved egg pickling (as shown in Figure 1).

## 2. Materials and Method

### 2.1. Materials

One-day-old duck eggs (70–75 g) were provided by Jiangsu Gaoyou Duck Development Group Co., Ltd., Yangzhou, China. Sodium hydroxide (NaOH), table salt (NaCl ≥ 98.5), urea, β-Mercaptoethanol (β-ME), potassium bromide (KBr), boric acid (H_3_BO_3_), glycerin, methyl red, bromocresol green, ethanol, plate count agar (PCA), glycine (Gly), ethylene diamine tetraacetic acid (EDTA), trimethylolaminomethane (Tris), 2-nitrobenzoic acid (DTNB), Coomassie Brilliant Blue G-250. All the other chemical reagents were purchased from Sinopharm Chemical Reagent Co., Ltd. (Shanghai, China). Deionized water (18.2 MΩ·cm) was used to prepare all aqueous solutions. The schematic diagram of the experimental flow is shown in Figure 1.

### 2.2. Sample Preparation

#### 2.2.1. Storage Experiment with Duck Eggs

One-day-old duck eggs were placed in a constant temperature and humidity control incubator of 25 °C and 4 °C, respectively (relative humidity was 40%); then, one part of the raw duck eggs with the number of tested days under 25 °C and 4 °C storage conditions was picked for freshness determination and alkali-induced EWG preparation, and the other parts were freeze-dried for molecular structure determination of EWP.

#### 2.2.2. Preparation of the Alkali-Induced EWG

The alkali-induced EWG was prepared according to the method described by Tan et al. [16], with some modifications. The duck eggs, which had been stored for the number of tested days, were manually broken and the albumen was separated from the yolk. The albumen solution was stirred magnetically at 200 r/min in an ice water bath for 2 h and then centrifuged at 8000 r/min for 15 min to remove the insoluble materials. Subsequently, a 20 g pre-treated albumen solution was mixed with 1 g NaOH solution (the final concentration of NaOH was 0.7%, *w*/*w*) and stirred at 200 r/min for a few minutes to let it stand to form a gel. Finally, the alkali-induced EWG was placed at room temperature for 3 days to determine its gelling properties.

### 2.3. Freshness Indicator Determination

#### 2.3.1. Aerobic Bacterial Count (ABC)

ABC was determined using the dilution plate method according to the Chinese National Standard GB 4789.2-2016 [17]. The counting unit is expressed in CFU. 

#### 2.3.2. Haugh Unit (HU) Measurement

The HU was employed to assess the freshness of eggs, as described by Shurmasti, D.K [18]. Each duck egg was first weighted and then broken horizontally, and the contents were poured onto a flat plate. Afterwards, the maximum height (mm) of albumen was determined to calculate the value of the HU according to Equation (1).
HU = 100 × log_10_ (*H* − 1.7*W*^0.37^ + 7.6)(1)
where H is the maximum height of the albumen (mm) and W is the mass of the whole shell egg (g).

#### 2.3.3. Albumen pH and Viscosity Measurement

The procedure involves the separation of duck egg whites from the duck eggs, followed by thorough mixing using a glass rod. Subsequently, the pH value of the albumin is directly measured using a STARTER 3100 pH meter (Aarhus Instruments Co., Ltd., Shanghai, China).

The viscosity of the albumen was determined by employing an NDJ-5S viscometer (Shanghai Lichen Bangxi Instrument Technology Co., Ltd., Shanghai, China). In this experiment, four eggs were selected as samples. The egg whites were carefully separated from the yolk. The egg white was then transferred into a 250 mL beaker and thoroughly mixed. Then, the #1 rotor was used and the speed was adjusted to 12 r/min. The viscosity of the egg white was measured using a viscometer at room temperature. The data were recorded subsequent to the attainment of steady values. The average value was taken for each sample following three measurements. 

#### 2.3.4. Moisture Contents of Albumen

The moisture contents of albumen were determined according to the Chinese National Standard (GB 5009.3-2016) [19]. 

#### 2.3.5. Air Chamber Height Measurement

Each duck egg was broken without damaging the air chamber, and the air chamber height was measured by a SANTO 8014 vernier caliper (Shanghai Saituo Hardware Tools Co., Ltd., Shanghai, China) from the inside of the egg (mm).

#### 2.3.6. T_2_ Relaxation Time of Albumin

The T_2_ relaxation time of the albumin was determined by an LF-NMR analyzer (NMI20-030 V-I, Niumag Co., Ltd., Suzhou, China), with minor modifications that were reported in our prior study [20]. T_21_, T_22_, and T_23_ were designed as the relaxation times of bound water, immobile water, and free water, respectively_,_ and A_21_, A_22_ and A_23_ were designed as the peak areas of bound water, immobile water, and free water, respectively. 

#### 2.3.7. Total Volatile Basic Nitrogen (TVB-N)

The TVB-N content was determined by the semi-micro nitrogen method in accordance with the Chinese National Standard (GB 5009.228-2016) [21]. 

### 2.4. Comprehensive Freshness Index (CFI) Calculations

The present study employed PCA and SRA to analyze nine single freshness indicators, namely HU, pH, air cell height, moisture contents, albumin viscosity, A_21_, A_22_, A_23_, and TVB-N. These indicators were utilized to calculate the CFI. PCA and SRA can reduce the dimensionality of the data and make the analysis simpler with a smaller set of variables [22]. The process of using PCA and SRA to establish the CFI of raw eggs is as follows:
(1)The load coefficient, which is the coefficient of new principal component data produced from the linear combination of the original data, connected the principal component with nine single freshness indicators, as shown in Equation (2):Pn = *Cnl* × *S*1+ *Cn*2 × *S*2 + *Cn*3 × *S*3 + *Cn*4 × *S*4 + …… + *Cni* × *Si*(2)
where P is the principal component, n is the number of principal components, C is the load coefficient, *S* is the single freshness indicator, and i is the number of single freshness indicators, which is 9 in this study; S1, S2, S3, S4, S5, S6, S7, S8, and S9 are HU, pH, air cell height, moisture contents, albumin viscosity, A_21_, A_22_, A_23_, and TVB-N, respectively.(2)The contribution rate is the proportion of each principal component calculated by calculating the proportion of eigenvalues, which are characteristic values of the covariance coefficient matrix. The function between the CFI and principal component was established by Equation (3).
CFI = *r*1 × *P*1 + *r*2 × *P*2 + …… + *rn* × *Pn*(3)
where *r* is the variance contribution rate, *P* is the principal component, and n is the number of principal components.

### 2.5. Determination of the Molecular Structure of EWP during Storage

#### 2.5.1. Secondary Structure 

The determination of the secondary structure of the EWP during storage was conducted using a Nicolet is50 FTIR spectrometer that was equipped with a single reflection diamond attenuated total reflectance (ATR) accessory (Thermo Electron Corporation, Massachusetts, USA). The spectral range analyzed was from 4000 to 400 cm^−1^. A total of 32 scans were conducted for each spectrum, with a resolution of 4 cm^−1^. The analysis of the secondary structures (α-helix, β-sheet, β-turn, and random coil) of the EWP was conducted within the spectral region of 1600~1700 cm^−1^ using PeakFit v4.12 (SeaSolve, Framigham, MA, USA).

#### 2.5.2. Intermolecular Forces

The method described by González-Guillén et al. [23] was employed to determine the changes in intermolecular forces between EWP during the storage period. A total of 10 mg of lyophilized albumin powder was homogenized with 0.5 mL of selective reagents (S1: 0.05 mol/L NaCl; S2: 0.6 mol/L NaCl; S3: 0.6 mol/L NaCl + 1.5 mol/L urea; S4: 0.6 mol/L NaCl + 8 mol/L urea; S5: 0.6 mol/L NaCl + 8 mol/L urea + 0.5 mol/L β-ME), which were prepared in 0.05 mol/L pH 7.0 phosphate buffer and then stirred at 4 °C for 1 h. The resulting solution was subsequently centrifuged at 10,000× *g* for 15 min. Ionic bonds (SA) were expressed as the soluble protein in S2 minus the soluble protein in S1; hydrogen bonds (SB) were expressed as the soluble protein in S3 minus the soluble protein in S2; hydrophobic interactions (SC) were expressed as the soluble protein in S4 minus the soluble protein in S3; and S–S bonds (SD) were expressed as the soluble protein in S5 minus the soluble protein in S4. The protein concentration in the supernatants was determined using the Coomassie Brilliant Blue method. The result of soluble protein contents was expressed as µg soluble protein/mL.

#### 2.5.3. Free Sulfhydryl (SH)

The free SH contents of the albumin during storage were determined by using Ellman’s reagent (DTNB solution) with minor modifications, which were reported in our prior study [20].

### 2.6. Textural Properties Analysis (TPA) of Alkali-Induced EWG

The TPA of the alkali-induced EWG was performed in an A-XT Plus textural analyzer (Stable Micro Systems, Godalming, UK) fitted with a flat plunger (model number: SMS–P/0.5). Alkali-induced EWG (10 mm in height and 20 mm in diameter) were compressed to 50% of their original height at pre-test speeds, test speeds, and post-test speeds of 2.0 mm/s with a trigger point load of 5 g [24]. 

### 2.7. Statistical Analysis

The experimental data are presented as the mean ± SD (three parallel experiments). A one-way ANOVA was used to identify significant differences (*p* < 0.05). Correlations between the freshness indexes or molecular structure of EWP and the gelling properties of alkali-induced EWG were determined using the Pearson correlation coefficient.

## 3. Results and Discussion

### 3.1. Freshness Indicator Analysis

The freshness of the raw eggs had an impact on the microbial quality and gelling properties of the preserved eggs, and the storage temperature and time had a significant impact on the freshness and microbial quality of the shell eggs. It can be seen from Table 1 that aerobic bacteria began to be detected after 16 days of storage at 25 °C and 28 days of storage at 4 °C. It may be because the activity of lysozyme in albumin decreased during long-term storage [25,26,27], which reduced their ability to resist external microorganisms. In addition, Tian et al. [28] found that the yields of preserved eggs without breakage produced using the eggs after 16 and 32 days of storage at 25 °C were low, at only 83.75% and 62.50%, respectively. According to the Chinese National Standard (GB/T9694-2014) [29], the breakage rate of preserved eggs is required to be ≤ 5%. Therefore, from the perspective of microbial quality and damage rate of preserved eggs, duck eggs stored at 25 °C and 4 °C for 16 and 28 days, respectively, were selected for freshness and alkali-induced EWG research.

The dynamic changes in HU, albumen pH, moisture contents of albumen, air chamber height, albumen viscosity, T2 relaxation time, and TVB-N value with the number of tested days under 25 °C and 4 °C storage conditions are shown in Table 2 and Table 3. As seen, the values of the HU, moisture contents of albumen, albumen viscosity, T_23_, A_21_, and A_23_ decreased with the increase in the number of days under 25 °C and 4 °C storage temperatures, and the values of albumen pH and air chamber height, TVB-N, T_22_, and A_22_ increased with the increase in the number of days under 25 °C and 4 °C storage temperatures, especially these freshness indicators, which changed rapidly at 25 °C storage temperature. 

The changes in these freshness indicators are mainly caused by the following two reasons. On the one hand, the water and CO_2_ inside the eggs will gradually transfer to the outside environment through the pores of the eggshells, which directly leads to the rise of albumen pH and the expansion of the air chamber. On the other hand, the rise in albumen pH would promote protein hydration and hydrolysis of the concentrated protein [7], thus changing the aggregation behavior of the proteins in the eggs. The findings of the study indicate that the freshness of eggshells is mostly influenced by storage temperature and duration [30]. In addition, according to the Chinese industry standard (SBT10638-2011) the eggshell still belongs to the AA grade within 25 °C for 12 days and 4 °C for 20 days [31]. However, it belongs to the A grade after 12 days of storage at 25 °C and 20 days of storage at 4 °C. 

### 3.2. CFI Calculations

In this study, the Pearson correlation coefficient was used to determine the correlation between the single freshness indicators of HU, pH, air cell height, moisture contents, albumin viscosity, A_21_, A_22_, A_23_, and TVB-N (Table 4). As seen, the Pearson correlation coefficient between the moisture contents and albumin pH was −0.627, indicating a negative and moderate correlation between these two indicators. The Pearson correlation coefficient between A_23_ and albumin pH was −0.683, indicating a negative but moderate correlation between these two indicators. Except for the correlation between the moisture contents and albumin pH and the correlation between A_23_ and albumin pH, the Pearson correlation coefficients between the other freshness indicators are all greater than 0.7, indicating that the HU, pH, air cell height, moisture contents, albumin viscosity, A_21_, A_22_, A_23_, and TVB-N were significantly correlated with each other (*p* < 0.05). Therefore, only one principal component has an eigenvalue greater than 1, and with a cumulative contribution rate of 90.598% due to the strong correlation between the freshness indicators, the results are shown in Table 5. Hence, as shown in Equation (4), the principal component had a load coefficient connection with nine single freshness indicators:*Y* = 0.313 *X*_1_ − 0.265 *X*_2_0.313 *X*_3_ + 0.284 *X*_4_ + 0.310 *X*_5_ + 0.300 *X*_6_ − 0.310 *X*_7_ + 0.303 *X*_8_ − 0.315 *X*_9_.(4)
where *Y* is the comprehensive freshness index obtained using the PCA method, *X*_1_ is HU, *X*_2_ is air chamber height, *X*_3_ is albumen pH, *X*_4_ is moisture contents, *X_5_* is albumin viscosity, *X*_6_ is A_21_, *X*_7_ is A_22_, *X*_8_ is A_23_, and *X*_9_ is TVB-N. 

Furthermore, the standardized data X_i_ should be substituted into the aforementioned formula in order to obtain the PCA score, as shown in Table 6.

The dependent variable in this study is the comprehensive score obtained from the PCA method. The independent variables consist of nine freshness indicators. The SRA method is employed to identify the indicators of TVB-N and moisture contents. A CFI is then established using the selected variables as new independent variables, as shown in Equation (5).
CFI = 1.231 × *TVB-N* + 0.750 × moisture content − 58.578.(5)
where CFI is the comprehensive freshness index obtained using PCA and SRA methods.

Furthermore, the formula provided above can be utilized to obtain the score of SRA by substituting the TVB-N and moisture contents with standardized data, as shown in Table 6. It is evident that under 25 °C storage conditions, the score of the principal component and stepwise regression showed a decreasing trend with the increase in storage days, indicating that the higher the score of the principal component and stepwise regression, the fresher the duck eggs. Furthermore, there was no significant difference between the scores and ranking of stepwise regression and principal components, indicating that the stepwise regression prediction model can be used to predict the freshness of the raw eggs.

Using PCA and SRA methods, the freshness data of duck eggs stored at 4 °C were also analyzed (Table 7, Table 8 and Table 9), and the CFI was obtained as CFI =1.156 × TVB-N + 0.522 × A_23_33.004.

### 3.3. Gelling Properties Analysis

The gelling properties of the alkali-induced EWG, which was prepared using albumin with the number of tested days under 25 °C and 4 °C storage conditions, were studied, as shown in Table 10. As seen, the springiness and resilience of alkali-induced EWG decreased gradually with the number of tested days under 25 °C and 4 °C storage conditions (*p* < 0.05), while its hardness and chewiness increased first and then decreased with the extension of storage time, especially the hardness, which reached its peak on the 12th and 20th days, and the chewiness, which reached its peak on the 8th day under 25 °C and 4 °C storage temperatures. The reason for this phenomenon may be related to the migration and loss of water as well as the conformational changes in EWP during storage (Table 2, Table 11). In addition, the decrease in bound water and the increase in immobilized water in albumin during storage (Table 3) promote the interaction between proteins and between proteins and water, thus conducive to the formation of a three-dimensional gel structure of EWG during alkali-induced conditions [32]. However, the hardness of alkali-induced EWG decreased significantly after 12 days of storage at 25 °C and 20 days of storage at 4 °C, but it is still higher than that of the one-day-old egg. It may be because the structure of the EWP becomes disorderly and loose, which is beneficial to the aggregation and reassembly of the EWP during alkali-induced conditions [9]. In addition, Carraro Alleoni [33] found that during storage, ovalbumin is altered to S-ovalbumin, which is extra heat-stable in comparison to ovalbumin. S-ovalbumin has a slightly lighter molecular weight than ovalbumin, and its relative quantity in the egg white can increase during the storage period, from 5% in fresh eggs to 81% after six months of refrigerated storage. S-ovalbumin is a protein with distinct and different abilities to form gel and foam. In the case of gels, its presence can diminish moisture loss and, along with other proteins, increase the hardness of albumen gel, a positive effect.

### 3.4. Correlation between CFI and Gelling Properties of Alkali-Induced EWG

In this study, a CFI was obtained using the PCA and SRA methods. As seen in Table 10, the hardness and chewiness of alkali-induced EWG first increased and then decreased with the number of tested days under 25 °C and 4 °C storage conditions, and the peak value appeared at 12 days at 25 °C and 20 days at 4 °C. Therefore, taking the time when the peak value of gelling properties appeared as the dividing point, the correlation between the CFI and the gelling properties of alkali-induced EWG before and after storage at 25 °C for 12 days and 4 °C for 20 days was analyzed (Table 12). 

As seen, stored at 25 °C for 12 days and at 4 °C for 20 days, the CFI of duck eggs showed a strong negative correlation with the hardness of alkali-induced EWG (*p* < 0.01), a high negative correlation with the chewiness (*p* < 0.05), and a strong positive correlation with the resilience (*p* < 0.01), indicating that the hardness and chewiness of alkali-induced EWG increased with the decrease in the CFI of duck eggs, while its resilience decreased with the decrease in the CFI of duck eggs within 12 days of storage at 25 °C and 20 days at 4 °C. In addition, the correlation between the TVB-N contents and the gelling properties of alkali-induced EWG is just opposite the CFI. The findings showed that the TVB-N value has a major impact on the CFI of duck eggs stored at 25 °C for 12 days and at 4 °C for 20 days and that change in the TVB-N value can also predict the gelling properties of alkali-induced EWG. Moreover, as shown in Table 12, after 12 days of storage at 25 °C, the CFI and moisture contents were significantly positively correlated with the hardness, springiness, and resilience of alkali-induced EWG (*p* < 0.01) and significantly positively correlated with the chewiness (*p* < 0.05), while the TVB-N contents showed a highly significant negative correlation with hardness, springiness, resilience, and chewiness (*p* < 0.01). The results showed that the gelling properties of alkali-induced EWG could be predicted by the CFI, moisture contents, and TVB-N contents after 12 days of storage at 25 °C. Similarly, after 20 days of storage at 4 °C, the correlation between the gelling properties of alkali-induced EWG and the CFI was consistent with that after 12 days of storage at 25 °C. Therefore, the gelling properties of alkali-induced EWG could be predicted by CFI, TVB-N, or A_23_ after 20 days of storage at 4 °C.

### 3.5. Structure Property Analysis

#### 3.5.1. Secondary Structure Analysis

The investigation of protein secondary structures has commonly utilized the amide I band, which falls within the spectral range of 1600~1700 cm^−1^ [34,35]. Moreover, 1650~1662 cm^−1^, 1610~1640 cm^−1^, 1662~1695 cm^−1^, and 1640~1650 cm^−1^ are considered as α-helix, β-sheet, β-turn and random coil structures, respectively. As shown in Table 7, the α-helix and β-turn of the EWP are significantly decreased, while the β-sheet and random coil contents showed significant increase trends as the storage time increased at 25 °C and 4 °C storage conditions (*p* < 0.05), consistent with the research results of Sheng et al. [36]. The decrease in α-helix contents and the increase in β-sheet and random coil contents during the storage process indicate that the molecule structure of the EWP undergoes uncoiling, which significantly increases the flexibility of the EWP. This is beneficial for the further denaturation, aggregation, and reassembly of EWP molecules during alkaline induction. In conclusion, the secondary structure of the EWP gradually changes from ordered to disordered with the extension of storage time, thereby enhancing the molecular flexibility of the EWP, which is consistent with the results of intermolecular forces (Table 11).

#### 3.5.2. Intermolecular Forces Analysis

As shown in Table 11, the main intermolecular forces between EWP in raw eggs are S–S bonds, followed by ionic bonds, hydrophobic interactions, and hydrogen bonds, which are consistent with the findings of Liu et al. [37]. In addition, the ionic bonds and S–S bonds significantly decreased with the number of tested days under 25 and 4 °C storage conditions, while hydrogen bonds significantly increased, and hydrophobic interactions decreased first and then increased with the extension of storage time under 25 and 4 °C storage conditions.

The reason for the decrease in ionic bonds may be due to the fact that the lysozyme is the only alkaline protein in egg whites (with an isoelectric point of 11.2); thus, its charge carrying capacity significantly decreased with the increase in albumin pH during storage at 25 °C and 4 °C (pH > 8), in turn reducing the ionic bonds between EWP. In addition, the reason for the changes in hydrophobic interactions may be that the hydrophobic amino acids are embedded in the internal structure of protein molecules in fresh raw eggs, but in the early stages of storage, the increase in β-sheet contents prevents the exposure of hydrophobic groups, resulting in a decrease in surface hydrophobicity. However, hydrophobic amino acids will gradually expose themselves to the surface of protein molecules as the storage time of raw eggs continues, leading to an increase in their hydrophobic interactions. In addition, the S–S bonds decrease with the extension of storage time under the storage conditions of 4 °C and 25 °C, further reducing the stability of the molecular conformation of EWP, which is consistent with the FTIR results. 

#### 3.5.3. Free SH Analysis

SH is the most active functional group in proteins, which can form S–S bonds under oxidation conditions, and the contents of free SH and S–S bonds are an important indicator to measure the change in protein structure and protein aggregation [38]. As seen in Table 6, the free SH significantly increased from 10.88 µmol/g to 13.20 µmol/g under storage at 25 °C, and the free SH first increased to 12.44 µmol/g from 10.88 µmol/g and then decreased to 11.25 µmol/g under storage at 4 °C. Moreover, the rate of change in free SH under 4 °C storage conditions is much lower than that under 25 °C storage, indicating that low temperatures can slow down the conformational changes in the proteins in eggs, reduce the changes in free SH in egg whites, and thus extend the shelf life of duck eggs. 

In addition, it can be seen from Table 11 that the free SH increases with the extension of storage days under storage conditions of 25 and 4 °C. It may be because during storage, the S–S bond in egg white protein undergoes a reduction reaction to form SH, and the decrease in S–S bond content can also prove this conclusion. The breaking of the S–S bond will form two SHs, and at the same time, the SH embedded inside the molecule will gradually be exposed on the surface of the protein molecule, thereby increasing the content of SH. However, there was a downward trend in free SH after storage at 4 °C for 20 days, which may be due to the oxidation of SH [39]. On the other hand, it may also be due to the formation of aggregates after protein denaturation, making it difficult to detect [40]. 

### 3.6. Correlations between Structure of EWP and Gelling Properties of Alkali-Induced EWG

This study also examined the relationship between the molecular structures of EWP and the gelling properties of alkali-induced EWG because the spatial conformation of EWP affects the gelling properties of alkali-induced EWG. 

As shown in Table 13, within 12 days of storage at 25 °C, the hardness of alkali-induced EWG was significantly negatively correlated with ionic bonds and S–S bonds (*p* < 0.01) and significantly positively correlated with hydrogen bonds (*p* < 0.01). On the other hand, resilience was significantly positively correlated with ionic bonds and S–S bonds (*p* < 0.01) and significantly negatively correlated with hydrogen bonds (*p* < 0.01). Springiness only had a significant negative correlation with the hydrogen bond (*p* < 0.05), while chewiness had a significant negative correlation with the α-helix and S–S bond (*p* < 0.05) and a highly significant negative correlation with hydrophobic interactions (*p* < 0.01). In addition, the correlation between the molecular structure of EWP and the gelling properties of alkali-induced EWG within 20 days of storage at 4 °C was consistent with that within 12 days of storage at 25 °C. The results indicate that the decrease in contents of S–S bonds and ionic bonds and the increase in hydrogen bonds are the main reasons for the increase in hardness and chewiness of alkali-induced EWG. Meanwhile, as shown in Table 8, the changes in elasticity and resilience are opposite to the changes in hardness and chewiness, indicating that EWP would undergo an optimal conformational transition within 12 days of storage at 25 °C and 20 days of storage at 4 °C, which could optimize the gelling properties of its alkali-induced EWG. 

However, after 12 days of storage at 25 °C or 20 days of storage at 4 °C, the gelling properties of alkali-induced EWG were significantly positively correlated with the contents of β-turn, ionic bonds, and S–S bonds (*p* < 0.05) and had a highly significant negative correlation with β-sheet and hydrogen bonds (*p* < 0.05). It indicates that when duck eggs are at level A, the decrease in α-helix, β-turn, ionic bonds, and S–S bonds and the increase in β-sheet and hydrogen bonds are the main reasons for the decrease in gelling properties of alkali-induced EWG. It is consistent with the research results of Tian et al. [28]. 

In summary, when the raw egg is stored at 25 °C for 12 days and 4 °C for 20 days, it still belongs to the AA level; thus, the decrease in the ion bonds, S–S bonds, and α-helix was beneficial to the increase in hardness of alkali-induced EWG. However, after storing the raw eggs at 25 °C for 12 days and 4 °C for 20 days, they still belong to the A level, and the decrease in ion bonds, S–S bonds, and β-turn will cause a decrease in the hardness, springiness, resilience, and chewiness of alkali-induced EWG. 

## 4. Conclusions

Using PCA and SRA methods, CFI = −1.23 X_1_+ 0.75 X_2_ − 58.58 and CFI = −1.16 X_1_ + 0.52 X_3_ − 33.00 (X_1_, TVB-N contents; X_2_, moisture content; X_3_, A_23_) can predict the freshness of duck eggs under storage conditions of 25 °C and 4 °C, respectively. TPA results indicated that the springiness and resilience of alkali-induced EWG decreased gradually with the number of tested days under 25 °C and 4 °C storage conditions (*p* < 0.05), while its hardness and chewiness increased first and then decreased with the extension of storage time, especially the hardness, which reached its peak on the 12th and 20th days, and the chewiness, which reached its peak on the 8th day under 25 °C and 4 °C storage temperatures. Correlation analysis demonstrated that the CFI showed a strong negative correlation with the hardness and chewiness of alkali-induced EWG and a strong positive correlation with resilience within 12 days of storage at 25 °C and 20 days at 4 °C (*p* < 0.01). Thus, the CFI can be used to predict the gelling properties of preserved eggs.

## Figures and Tables

**Figure 1 foods-12-04028-f001:**
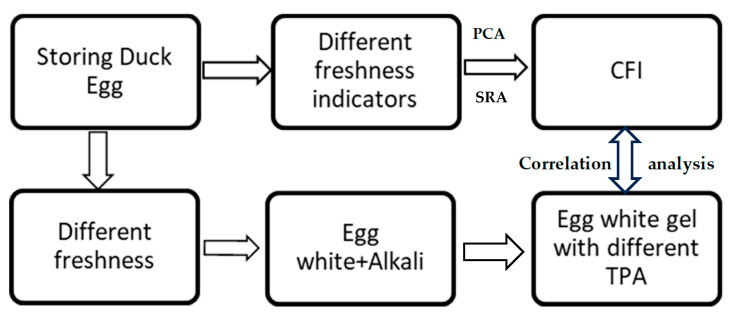
Schematic diagram of experimental flow.

**Table 1 foods-12-04028-t001:** Total number of aerobic bacterial in eggs stored at 25 °C and 4 °C.

Storage Time (Days)	Aerobic Bacterial Count (CFU)		
	Undiluted	10^−1^	10^−2^
Storage at 25 °C			
0	0	0	0
2	0	0	0
4	0	0	0
6	0	0	0
8	0	0	0
10	0	0	0
12	0	0	0
14	0	0	0
16	1	0	0
Storage at 4 °C			
0	0	0	0
4	0	0	0
8	0	0	0
12	0	0	0
16	0	0	0
20	0	0	0
24	0	0	0
28	1	0	0

**Table 2 foods-12-04028-t002:** Changes in the freshness indicator of a duck egg during storage at 25 °C and 4 °C.

Storage Time (Days)	Freshness Indicator					
	Haugh Unit	Albumen pH	Moisture Content (%)	Air Chamber Height (mm)	Viscosity (mPa·s)	TVB-N (mg/100 g)
Storage at 25 °C						
0	88.12 ± 1.59 ^a^	8.76 ± 0.02 ^f^	87.13 ± 0.17 ^a^	0.85 ± 0.08 ^i^	226.00 ± 4.20 ^a^	2.31 ± 0.30 ^g^
2	82.37 ± 0.79 ^b^	8.97 ± 0.13 ^e^	86.84 ± 0.23 ^b^	2.46 ± 0.07 ^h^	207.00 ± 4.24 ^b^	3.22 ± 0.40 ^f^
4	76.58 ± 2.13 ^c^	9.15 ± 0.06 ^d^	86.26 ± 0.09 ^de^	3.78 ± 0.19 ^g^	166.50 ± 4.95 ^c^	3.85 ± 0.10 ^e^
6	76.69 ± 2.38 ^c^	9.37 ± 0.08 ^c^	86.69 ± 0.13 ^bc^	4.66 ± 0.26 ^f^	134.50 ± 4.95 ^d^	4.34 ± 0.20 ^e^
8	75.83 ± 1.61 ^cd^	9.42 ± 0.04 ^c^	86.49 ± 0.09 ^cd^	5.44 ± 0.20 ^e^	123.50 ± 4.95 ^e^	5.11 ± 0.30 ^d^
10	73.13 ± 1.51 ^d^	9.64 ± 0.04 ^a^	86.30 ± 0.22 ^de^	6.17 ± 0.04 ^d^	114.51 ± 3.54 ^ef^	5.67 ± 0.11 ^cd^
12	69.11 ± 2.38 ^e^	9.53 ± 0.03 ^b^	86.47 ± 0.02 ^cd^	6.63 ± 0.26 ^c^	106.51 ± 2.12 ^f^	6.09 ± 0.30 ^c^
14	63.41 ± 1.54 ^f^	9.54 ± 0.02 ^ab^	86.17 ± 0.03 ^e^	7.73 ± 0.26 ^b^	86.81 ± 5.10 ^g^	7.07 ± 0.10 ^b^
16	56.83 ± 2.25 ^g^	9.48 ± 0.03 ^bc^	85.31 ± 0.10 ^f^	8.80 ± 0.25 ^a^	51.40 ± 1.84 ^h^	7.91 ± 0.30 ^a^
Storage at 4 °C						
0	88.12 ± 1.59 ^a^	8.76 ± 0.02 ^c^	87.13 ± 0.17 ^a^	0.85 ± 0.08 ^h^	226.00 ± 4.24 ^a^	2.31 ± 0.30 ^h^
4	86.89 ± 2.68 ^a^	8.92 ± 0.23 ^b^	86.46 ± 0.14 ^b^	2.45 ± 0.03 ^g^	186.00 ± 4.24 ^b^	2.94 ± 0.20 ^g^
8	81.17 ± 2.94 ^b^	9.13 ± 0.03 ^ab^	86.49 ± 0.09 ^b^	3.95 ± 0.24 ^f^	173.50 ± 3.54 ^c^	3.71 ± 0.30 ^f^
12	81.49 ± 0.48 ^b^	9.23 ± 0.02 ^a^	86.26 ± 0.36 ^b^	5.03 ± 0.37 ^e^	151.50 ± 4.95 ^d^	4.27 ± 0.10 ^e^
16	76.76 ± 1.37 ^c^	9.29 ± 0.06 ^a^	86.49 ± 0.05 ^b^	5.88 ± 0.02 ^d^	133.50 ± 3.54 ^e^	4.83 ± 0.10 ^d^
20	75.91 ± 1.43 ^c^	9.24 ± 0.02 ^a^	86.2 ± 0.04 ^b^	7.16 ± 0.16 ^c^	124.50 ± 2.12 ^f^	5.53 ± 0.31 ^c^
24	66.40 ± 1.55 ^d^	9.20 ± 0.03 ^a^	85.65 ± 0.2 ^c^	7.57 ± 0.12 ^b^	111.00 ± 1.41 ^g^	6.23 ± 0.11 ^b^
28	62.46 ± 0.92 ^e^	9.02 ± 0.01 ^bc^	85.71 ± 0.22 ^c^	8.76 ± 0.05 ^a^	98.35 ± 5.16 ^h^	7.01 ± 0.21 ^a^

Different superscript letters in the table indicate a significant difference (*p* < 0.05).

**Table 3 foods-12-04028-t003:** Relaxation time and peak area of duck egg white during storage at 25 °C and 4 °C.

Storage Time (days)	T_21_ (ms)	T_22_ (ms)	T_23_ (ms)	A_21_ (%)	A_22_ (%)	A_23_ (%)
Storage at 25 °C						
0	0.69 ± 0.00 ^a^	7.93 ± 0.00 ^b^	464.16 ± 0.00 ^a^	22.39 ± 0.35 ^a^	1.75 ± 0.21 ^f^	75.48 ± 0.16 ^a^
2	0.69 ± 0.00 ^a^	7.93 ± 0.00 ^b^	394.42 ± 0.00 ^b^	22.47 ± 0.43 ^a^	2.59 ± 0.28 ^e^	74.50 ± 0.52 ^ab^
4	0.69 ± 0.00 ^a^	8.39 ± 0.81 ^b^	394.42 ± 0.00 ^b^	22.31 ± 0.24 ^a^	3.12 ± 0.45 ^e^	74.49 ± 0.39 ^ab^
6	0.69 ± 0.00 ^a^	8.39 ± 0.81 ^b^	394.42 ± 0.00 ^b^	22.29 ± 0.70 ^a^	3.95 ± 0.31 ^d^	74.08 ± 0.02 ^bc^
8	0.69 ± 0.00 ^a^	8.39 ± 0.81 ^b^	335.16 ± 0.00 ^c^	20.97 ± 0.21 ^ab^	4.52 ± 0.38 ^d^	74.62 ± 1.17 ^ab^
10	0.69 ± 0.00 ^a^	8.86 ± 0.81 ^ab^	335.16 ± 0.00 ^c^	20.45 ± 0.66 ^bc^	6.56 ± 0.33 ^c^	73.04 ± 0.52 ^cd^
12	0.69 ± 0.00 ^a^	8.86 ± 0.81 ^ab^	335.16 ± 0.00 ^c^	19.21 ± 1.42 ^c^	8.36 ± 0.47 ^b^	72.04 ± 0.40 ^de^
14	0.69 ± 0.00 ^a^	9.41 ± 1.53 ^ab^	284.80 ± 0.00 ^d^	19.44 ± 1.30 ^c^	8.81 ± 0.35 ^b^	71.69 ± 0.33 ^e^
16	0.69 ± 0.00 ^a^	10.43 ± 0.95 ^a^	284.80 ± 0.00 ^d^	17.69 ± 0.35 ^d^	12.51 ± 0.39 ^a^	69.81 ± 1.52 ^f^
Storage at 4 °C						
4	0.69 ± 0.00 ^a^	7.93 ± 0.00 ^b^	464.16 ± 0.00 ^a^	22.26 ± 2.00 ^a^	2.59 ± 0.46 ^f^	74.43 ± 2.06 ^ab^
8	0.69 ± 0.00 ^a^	8.39 ± 0.81 ^ab^	394.42 ± 0.00 ^b^	21.67 ± 1.54 ^ab^	3.28 ± 0.22 ^e^	73.67 ± 0.29 ^b^
12	0.69 ± 0.00 ^a^	8.39 ± 0.81 ^ab^	394.42 ± 0.00 ^b^	21.21 ± 0.41 ^ab^	4.54 ± 0.33 ^d^	72.99 ± 0.11 ^bc^
16	0.69 ± 0.00 ^a^	8.39 ± 0.81 ^ab^	394.42 ± 0.00 ^b^	21.24 ± 0.69 ^ab^	5.04 ± 0.43 ^d^	73.33 ± 1.00 ^b^
20	0.69 ± 0.00 ^a^	8.86 ± 0.81 ^ab^	394.42 ± 0.00 ^b^	20.83 ± 0.28 ^ab^	6.12 ± 0.26 ^c^	73.09 ± 0.40 ^bc^
24	0.69 ± 0.00 ^a^	8.86 ± 0.81 ^ab^	335.16 ± 0.00 ^c^	19.82 ± 0.82 ^bc^	7.58 ± 0.09 ^b^	72.72 ± 0.79 ^bc^
28	0.69 ± 0.00 ^a^	9.34 ± 0.00 ^a^	335.16 ± 0.00 ^c^	18.57 ± 0.62 ^c^	9.58 ± 0.38 ^a^	71.55 ± 0.16 ^c^

Different superscript letters in the table indicate a significant difference (*p* < 0.05).

**Table 4 foods-12-04028-t004:** Correlation of single index of freshness at 25 °C.

	HU	pH	Air Chamber Height	Moisture Contents	Viscosity	A_21_	A_22_	A_23_	TVB-N
HU	1.000								
pH	−0.776 *	1.000							
Air chamber height	−0.970 **	0.900 **	1.000						
Moisture contents	0.916 **	−0.627	−0.849 **	1.000					
Viscosity	0.954 **	−0.900 **	−0.989 **	0.849 **	1.000				
A_21_	0.919 **	−0.704 *	−0.932 **	0.812 **	0.885 **	1.000			
A_22_	−0.965 **	0.735 *	0.957 **	−0.866 **	−0.922 **	−0.981 **	1.000		
A_23_	0.958 **	−0.683 *	−0.919 **	0.862 **	0.881 **	0.945 **	−0.986 **	1.000	
TVB-N	−0.979 **	0.853 **	0.996 **	−0.860 **	−0.977 **	−0.942 **	0.967 **	−0.937 **	1.000

** Correlation is significant at the 0.01 level (2-tailed), (*p* < 0.01). * Correlation is significant at the 0.05 level (2-tailed), (*p* < 0.05).

**Table 5 foods-12-04028-t005:** Statistical table of principal component information at 25 °C.

Principal Component	Eigenvalue	Variance Contribution Rate (%)	Cumulative (%)
1	8.154	90.598	90.598
2	0.509	5.659	96.257
3	0.226	2.515	98.772
4	0.056	0.618	99.390
5	0.037	0.411	99.801
6	0.014	0.159	99.960
7	0.003	0.035	99.995
8	0.000	0.005	100.000
9	5.336 × 10^−17^	5.928 × 10^−16^	100.000

**Table 6 foods-12-04028-t006:** Composite scores of principal components and stepwise regression scores at 25 °C.

Storage Time (days)	Score of PCA	Ranking	Score of SRA	Ranking
0	3.889	1	3.926	1
2	2.664	2	2.588	2
4	1.417	3	1.378	3
6	0.947	4	1.097	4
8	0.283	5	−0.001	5
10	−0.878	6	−0.833	6
12	−1.531	7	−1.222	7
14	−2.368	8	−2.654	8
16	−4.423	9	−4.333	9

**Table 7 foods-12-04028-t007:** Correlation of a single index of freshness at 4 °C.

	HU	pH	Air Chamber Height	Moisture Contents	Viscosity	A_21_	A_22_	A_23_	TVB-N
HU	1.000								
pH	−0.420	1.000							
air chamber height	−0.950 **	0.654	1.000						
moisture contents	0.919 **	−0.380	−0.870 **	1.000					
viscosity	0.918 **	−0.688	−0.977 **	0.869 **	1.000				
A_21_	0.977 **	−0.337	−0.933 **	0.919 **	0.890 **	1.000			
A_22_	−0.980 **	0.437	0.973 **	−0.923 **	−0.944 **	−0.986 **	1.000		
A_23_	0.896 **	−0.605	−0.933 **	0.862 **	0.949 **	0.919 **	−0.932 **	1.000	
TVB-N	−0.973 **	0.553	0.994 **	−0.907 **	−0.976 **	−0.957 **	0.989 **	−0.939 **	1.000

** Correlation is significant at the 0.01 level (2-tailed), (*p* < 0.01).

**Table 8 foods-12-04028-t008:** Statistical table of principal component information at 4 °C.

Principal Component	Eigenvalue	Variance Contribution Rate (%)	Cumulative (%)
1	7.866	87.396	87.396
2	0.873	9.700	97.096
3	0.123	1.371	98.467
4	0.089	0.989	99.456
5	0.035	0.390	99.846
6	0.010	0.112	99.958
7	0.004	0.042	100.000
8	1.221 × 10^−15^	1.356 × 10^−14^	100.000
9	−1.055 × 10^−16^	−1.172 × 10^−15^	100.000

**Table 9 foods-12-04028-t009:** Composite scores of principal components and stepwise regression scores at 4 °C.

Storage Time (days)	Score of PCA	Ranking	Score of SRA	Ranking
0	3.798	1	3.726	1
4	2.375	2	2.450	2
8	1.172	3	1.163	3
12	0.102	4	0.161	4
16	−0.244	5	−0.309	5
20	−1.062	6	−1.244	6
24	−2.518	7	−2.246	7
28	−3.624	8	−3.747	8

**Table 10 foods-12-04028-t010:** Changes in texture properties of alkali-induced egg white protein gel.

Storage Time (days)	Hardness (g)	Springness (%)	Resilience (%)	Chewiness (g·mm)
25 °C storage				
0	452.33 ± 4.84 ^f^	23.14 ± 0.34 ^a^	18.71 ± 0.24 ^a^	3714.14 ± 98.93 ^e^
2	464.37 ± 3.17 ^ef^	23.03 ± 0.07 ^ab^	17.97 ± 0.25 ^b^	3906.04 ± 125.70 ^d^
4	491.45 ± 47.36 ^de^	22.60 ± 0.37 ^ab^	18.06 ± 0.15 ^b^	3973.93 ± 223.61 ^d^
6	501.34 ± 12.77 ^d^	22.47 ± 0.82 ^ab^	17.45 ± 0.29 ^c^	4586.36 ± 37.00 ^b^
8	579.92 ± 3.81 ^bc^	22.39 ± 0.12 ^b^	16.84 ± 0.15 ^d^	4857.95 ± 32.96 ^a^
10	594.93 ± 13.16 ^b^	22.61 ± 0.44 ^ab^	16.59 ± 0.20 ^d^	4903.08 ± 85.25 ^a^
12	664.24 ± 5.78 ^a^	20.18 ± 0.19 ^c^	15.74 ± 0.28 ^e^	4303.10 ± 10.91 ^c^
14	559.10 ± 18.52 ^c^	18.34 ± 0.08 ^d^	14.25 ± 0.15 ^f^	2943.94 ± 69.24 ^f^
16	487.97 ± 10.40 ^de^	17.03 ± 0.19 ^e^	12.70 ± 0.51 ^g^	2695.85 ± 75.04 ^g^
4 °C storage				
4	571.45 ± 19.63 ^e^	22.66 ± 0.18 ^a^	18.36 ± 0.25 ^ab^	4163.65 ± 48.36 ^b^
8	613.07 ± 19.41 ^d^	20.79 ± 0.37 ^b^	17.84 ± 0.15 ^b^	5603.99 ± 16.75 ^a^
12	662.92 ± 11.06 ^c^	20.84 ± 0.26 ^b^	16.35 ± 0.17 ^c^	3730.23 ± 99.02 ^c^
16	693.06 ± 7.54 ^b^	18.26 ± 0.18 ^cd^	14.78 ± 0.16 ^d^	3615.85 ± 43.88 ^d^
20	722.78 ± 14.11 ^a^	18.64 ± 0.05 ^c^	14.04 ± 0.45 ^de^	2849.88 ± 27.43 ^e^
24	632.47 ± 3.65 ^d^	18.59 ± 0.50 ^c^	13.29 ± 0.65 ^e^	2682.77 ± 13.42 ^f^
28	470.77 ± 9.38 ^f^	17.87 ± 0.31 ^d^	11.57 ± 0.90 ^f^	2433.52 ± 9.08 ^g^

Different superscript letters in the table indicate a significant difference (*p* < 0.05).

**Table 11 foods-12-04028-t011:** Changes in secondary structure of egg white protein during storage at 25 °C.

Storage Time (days)	α-helix (%)	β-sheet (%)	β-turn (%)	Random Coil(%)	SA (µg/mL)	SB (µg/mL)	SC (µg/mL)	SD (µg/mL)	Free SH (µmol/g)
Storage at 25 °C									
0	23.66 ± 0.21 ^a^	36.60 ± 0.27 ^c^	27.11 ± 0.22 ^a^	12.79 ± 0.01 ^b^	211.25 ± 1.77 ^a^	132.50 ± 7.07 ^e^	196.25 ± 1.77 ^a^	488.00 ± 4.24 ^a^	10.88 ± 0.31 ^d^
2	23.24 ± 0.19 ^ab^	36.59 ± 0.33 ^c^	23.51 ± 3.93 ^b^	13.28 ± 0.98 ^b^	205.00 ± 3.54 ^ab^	131.25 ± 1.77 ^e^	190.00 ± 3.54 ^bc^	486.25 ± 1.77 ^a^	11.99 ± 0.10 ^c^
4	22.78 ± 0.08 ^bc^	41.07 ± 0.41 ^b^	21.95 ± 0.46 ^b^	13.93 ± 0.01 ^ab^	201.25 ± 8.84 ^bc^	133.50 ± 5.66 ^e^	187.75 ± 0.35 ^de^	485.75 ± 2.47 ^a^	12.58 ± 0.10 ^b^
6	22.88 ± 0.43 ^bc^	40.97 ± 0.66 ^b^	22.12 ± 0.21 ^b^	14.33 ± 0.08 ^ab^	198.50 ± 2.12 ^bc^	139.00 ± 2.12 ^de^	185.00 ± 3.54 ^de^	478.50 ± 1.41 ^b^	12.53 ± 0.34 ^bc^
8	22.84 ± 0.03 ^bc^	41.64 ± 0.39 ^b^	21.64 ± 0.22 ^b^	13.96 ± 0.18 ^ab^	193.75 ± 5.30 ^cd^	147.50 ± 3.53 ^cd^	183.75 ± 1.77 ^de^	475.75 ± 4.60 ^b^	12.81 ± 0.16 ^ab^
10	22.36 ± 0.66 ^c^	41.37 ± 0.18 ^b^	21.73 ± 0.04 ^b^	14.46 ± 0.33 ^ab^	183.75 ± 5.30 ^de^	151.50 ± 4.95 ^bc^	182.25 ± 0.35 ^e^	475.00 ± 3.54 ^b^	12.77 ± 0.10 ^ab^
12	22.80 ± 0.03 ^bc^	41.50 ± 0.10 ^b^	21.59 ± 0.23 ^b^	14.25 ± 0.12 ^ab^	183.50 ± 1.41 ^de^	160.50 ± 7.07 ^ab^	188.50 ± 1.41 ^cd^	471.75 ± 4.60 ^bc^	13.03 ± 0 ^ab^
14	22.66 ± 0.14 ^c^	41.47 ± 0.62 ^b^	21.63 ± 0.48 ^b^	13.96 ± 0.17 ^ab^	183.75 ± 1.77 ^de^	166.25 ± 1.77 ^a^	197.50 ± 3.54 ^a^	471.00 ± 1.41 ^bc^	13.31 ± 0.45 ^a^
16	20.90 ± 0.07 ^d^	43.94 ± 0.26 ^a^	21.26 ± 0.45 ^b^	15.33 ± 2.49 ^a^	174.75 ± 2.47 ^e^	167.75 ± 2.47 ^a^	193.75 ± 1.77 ^ab^	466.25 ± 1.77 ^c^	12.73 ± 0.21 ^ab^
Storage at 4 °C									
4	23.43 ± 0.46 ^ab^	36.14 ± 0.10 ^f^	22.17 ± 0.01 ^b^	14.40 ± 0.44 ^a^	206.50 ± 2.12 ^ab^	135.50 ± 0.71 ^e^	190.50 ± 1.41 ^ab^	487.50 ± 2.83 ^a^	10.88 ± 0.31 ^b^
8	22.93 ± 0.03 ^bc^	40.49 ± 0.26 ^cd^	22.06 ± 0.06 ^b^	14.38 ± 0.28 ^a^	203.75 ± 1.77 ^bc^	138.75 ± 1.77 ^de^	188.00 ± 1.41 ^bc^	485.50 ± 4.24 ^a^	12.07 ± 0.21 ^a^
12	23.03 ± 0.07 ^bc^	40.82 ± 0.13 ^c^	22.23 ± 0.09 ^b^	14.22 ± 0.01 ^a^	200.75 ± 4.60 ^bc^	144.25 ± 1.06 ^cd^	186.00 ± 1.41 ^bcd^	481.50 ± 5.66 ^ab^	12.40 ± 0.16 ^a^
16	22.74 ± 0.12 ^cd^	40.31 ± 0.26 ^d^	21.49 ± 0.18 ^cd^	14.38 ± 0.35 ^a^	198.75 ± 1.77 ^c^	146.50 ± 1.41 ^bcd^	179.75 ± 0.35 ^de^	475.25 ± 3.18 ^bc^	12.03 ± 0.16 ^a^
20	22.24 ± 0.64 ^d^	41.38 ± 0.18 ^b^	21.64 ± 0.11 ^c^	14.31 ± 0.25 ^a^	190.50 ± 2.83 ^d^	149.75 ± 3.89 ^bc^	176.10 ± 4.10 ^e^	473.65 ± 4.45 ^bc^	12.10 ± 0.05 ^a^
24	23.01 ± 0.03 ^bc^	41.34 ± 0.20 ^b^	21.76 ± 0.19 ^c^	14.39 ± 0.08 ^a^	182.50 ± 3.54 ^e^	154.00 ± 4.95 ^ab^	183.50 ± 5.66 ^cd^	466.75 ± 1.06 ^cd^	12.44 ± 0.21 ^a^
28	22.68 ± 0.18 ^cd^	42.37 ± 0.47 ^a^	21.21 ± 0.24 ^d^	14.21 ± 0.22 ^a^	173.75 ± 1.77 ^f^	159.85 ± 0.21 ^a^	186.65 ± 1.63 ^bc^	461.00 ± 4.95 ^d^	11.21 ± 0.05 ^b^

Different superscript letters in the table indicate a significant difference (*p* < 0.05).

**Table 12 foods-12-04028-t012:** Correlation analysis between physicochemical indexes of freshness and alkali-induced gel texture properties.

	Hardness	Springness	Resilience	Chewiness
**Within 12 days of storage at 25 °C**				
CFI	−0.940 **	0.745	0.959 **	−0.743
TVB-N	0.950 **	−0.728	−0.972 **	0.781 *
Moisture contents	−0.604	0.378	0.597	−0.575
**Storage at 25 °C for more than 12 days**				
CFI	0.954 **	0.932 **	0.985 **	0.847 *
TVB-N	−0.994 **	−0.988 **	−0.965 **	−0.955 **
Moisture contents	0.942 **	0.921 **	0.969 **	0.835 *
**Within 20 days of storage at 4 °C**				
CFI	−0.989 **	0.923 **	0.925 **	0.338
TVB-N	0.964 **	−0.950 **	−0.967 **	−0.425
A_23_	−0.961 **	0.814 *	0.786	0.145
**Storage at 4 °C for more than 20 days**				
CFI	0.981 **	0.879 **	0.974 **	0.986 **
TVB-N	−0.992 **	−0.904 **	−0.987 **	−0.995 **
A_23_	0.974 **	0.961 **	0.996 **	0.984 **

** Correlation is significant at the 0.01 level (2-tailed), (*p* < 0.01). * Correlation is significant at the 0.05 level (2-tailed), (*p* < 0.05).

**Table 13 foods-12-04028-t013:** Correlation analysis between molecular structure and alkali-induced gel texture properties of duck egg white.

	α-Helix	β-Sheet	β-Turn	Random Coil	SA	SB	SC	SD	Free SH
**Within 12 days of storage at 25 °C**									
Hardness	−0.668	0.726	−0.653	0.700	−0.945 **	0.990 **	−0.534	−0.940 **	0.763 *
Springness	0.356	−0.538	0.490	−0.507	0.698	−0.820 *	0.166	0.733	−0.609
Resilience	0.708	−0.723	0.735	−0.769 *	0.958 **	−0.966 **	0.620	0.967 **	−0.825 *
Chewiness	−0.780 *	0.760 *	−0.701	0.797 *	−0.746	0.643	−0.911 **	−0.809 *	0.710
**Storage at 25 °C for more than 12 days**									
Hardness	0.916 **	−0.861 *	0.851 *	−0.762 *	0.856 *	−0.967 **	−0.395	0.901 **	0.668
Springness	0.881 **	−0.828 *	0.805 *	−0.706	0.813 *	−0.986 **	−0.468	0.868 *	0.610
Resilience	0.952 **	−0.929 **	0.933 **	−0.854 *	0.938 **	−0.898 **	−0.212	0.964 **	0.780 *
Chewiness	0.787 *	−0.709	0.690	−0.573	0.695	−0.997 **	−0.626	0.762 *	0.454
**Within 20 days of storage at 4 °C**									
Hardness	−0.916 *	0.843 *	−0.881 *	0.801	−0.931 **	0.956 **	−0.955 **	−0.873 *	0.842 *
Springness	0.922 **	−0.837 *	0.688	−0.578	0.885 *	−0.935 **	0.954 **	0.952 **	−0.652
Resilience	0.910 *	−0.774	0.603	−0.475	0.942 **	−0.976 **	0.966 **	0.997 **	−0.551
Chewiness	0.319	−0.099	−0.054	0.170	0.481	−0.490	0.443	0.599	0.160
**Storage at 4 °C for more than 20 days**									
Hardness	0.131	−0.923 **	0.891 **	−0.233	0.987 **	−0.990 **	−0.946 **	0.978 **	0.797 *
Springness	0.508	−0.983 **	0.964 **	−0.508	0.918 **	−0.888 **	−0.815 *	0.883 **	0.553
Resilience	0.261	−0.964 **	0.941 **	−0.305	0.990 **	−0.985 **	−0.922 **	0.972 **	0.730
Chewiness	0.164	−0.933 **	0.904 **	−0.239	0.994 **	−0.995 **	−0.948 **	0.984 **	0.781 *

** Correlation is significant at the 0.01 level (2-tailed), (*p* < 0.01). * Correlation is significant at the 0.05 level (2-tailed), (*p* < 0.05).

## Data Availability

The data used to support the findings of this study can be made available by the corresponding author upon request.

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
