# Peer review of "Predict the Gelling Properties of Alkali-Induced Egg White Gel Based on the Freshness of Duck Eggs"

_foods, 2023, doi:10.3390/foods12214028_

Round 1
Reviewer 1 Report
Comments and Suggestions for Authors
Dear authors,
1. The topic is very interesting, creative and practical and the manuscript contains very interesting information.
2. Various and extensive tests have been conducted along with accurate statistical analysis.
3. The manuscript title is very general. This is while only the relationship between freshness indicators and gelling properties has been evaluated and maybe the general term "egg white quality" cannot be used. Other functional properties of egg white, such as foaming, emulsifying, coating and nutritional properties, have not been measured.
4. Considering the nature of the research and various tests, the abstract of the manuscript is not comprehensive enough and does not provide a clear understanding of the dimensions of the research. In my opinion, it is not interesting to include the CFI equation in the abstract. I recommend that the abstract be rewritten realistically.
5. Can the improvement of gelling properties of egg white during storage be related to the formation of S-ovalbumin? In some scientific references, it is mentioned: “During storage, ovalbumin is altered to s-ovalbumin, an extra heat-stable in comparison to ovalbumin. S-ovalbumin has a slightly lighter molecular weight than ovalbumin and its relative quantity in the eggwhite can increase during the storage period, from 5% in fresh eggs to 81% after six months of refrigerated storage. The s-ovalbumin is a protein with distinct, different ability to form gel and foam. In the case of gels, its presence can diminish the moisture loss, a positive effect.”(https://www.scielo.br/j/sa/a/7W4n5V7gP9CPGWBGXMWQPBq/)
6. In the introduction section, before the last paragraph, it is better to mention the latest achievements in relation to predicting egg white quality based on freshness indicators. This can clarify the necessity of the research and its objectives and innovation.
7. Due to the large number of specialized and abbreviations terms, it's good that add a table of symbols and abbreviations to the manuscript. For example, up to page 8 of the manuscript, despite referring several times to parameters such as A21, A22, etc., there is no information about their nature.
8. In the materials and methods section, it is better to show the schematic flow diagram for experiments.
9. It is better to arrange Figure 1 in a regular arrangement and in the form of one figure and one caption. In the current state, the titles are duplicated and it has caused congestion. Also, Table 1. A and Table 1. B do not need to be separated.
10. According to the results of parameters such as pH, moisture content and viscosity, is the beginning of detection of aerobic bacteria only related to lysozyme activity?
11. Lines 212-227: It is better to discuss the reasons and mechanism of changes of freshness indicators during storage more (not necessarily in detail).
12. Table 3.A and 4.A: Please check the significance or non-significance of the correlation between freshness indicators and inserting one or two stars.
13. What is Figure 5A in lines 308-309? Do you mean Table 5A?
14. Please introduce terms CSF and CSFM in Table 5.B.
15. I think the gelling behavior of egg white during storage has not been properly discussed in terms of the effect of temperature and time. Apart from the correlation between freshness indicators and gelling properties, the mechanisms are not clear.
16. I could not find Figure 26 on page 15 (line 339).
17. The difference between abstract and conclusion is not respected and similar sentences are used. I recommend that according to the good results obtained in this research, the conclusion should be revised and the achievements reflected.
Author Response
Dear editor and reviewers,
Thank you very much for your constructive and valuable suggestions. We have made careful modifications to the revised manuscript of the first edition. All changes made to the text are in red. Our responses to several comments are listed below:
Response to reviewer # 1
Comment (1):
The manuscript title is very general. This is while only the relationship between freshness indicators and gelling properties has been evaluated and maybe the general term "egg white quality" cannot be used. Other functional properties of egg white, such as foaming, emulsifying, coating and nutritional properties, have not been measured.
Response (1): Thank you for your suggestion. “Predict the gelling properties of alkali-induced egg white gel based on the freshness of duck eggs” was used to replace “Predict the quality of the preserved egg white based on the freshness of shell eggs”.
Comment (2):
Considering the nature of the research and various tests, the abstract of the manuscript is not comprehensive enough and does not provide a clear understanding of the dimensions of the research. In my opinion, it is not interesting to include the CFI equation in the abstract. I recommend that the abstract be rewritten realistically.
Response (2): Thank you. The abstract has been improved in the revised manuscript according to your suggestion.
Comment (3):
Can the improvement of gelling properties of egg white during storage be related to the formation of S-ovalbumin? In some scientific references, it is mentioned: “During storage, ovalbumin is altered to s-ovalbumin, an extra heat-stable in comparison to ovalbumin. S-ovalbumin has a slightly lighter molecular weight than ovalbumin and its relative quantity in the egg white can increase during the storage period, from 5% in fresh eggs to 81% after six months of refrigerated storage. The s-ovalbumin is a protein with distinct, different ability to form gel and foam. In the case of gels, its presence can diminish the moisture loss, a positive effect.” (https://www.scielo.br/j/sa/a/7W4n5V7gP9CPGWBGXMWQPBq/)
Response (3): Thank you for your suggestion. We have added an explanation about the improvement in gel properties related to S-ovalbumin formation in “Section 3.3 Gelling properties analysis”. Details can be found in lines 314-321.
Comment (4):
In the introduction section, before the last paragraph, it is better to mention the latest achievements in relation to predicting egg white quality based on freshness indicators. This can clarify the necessity of the research and its objectives and innovation.
Response (4): Thank you. We have added to the introduction about the latest achievements in relation to predicting egg white quality based on freshness indicators. Details can be found in lines 57-65.
Comment (5):
Due to the large number of specialized and abbreviations terms, it's good that add a table of symbols and abbreviations to the manuscript. For example, up to page 8 of the manuscript, despite referring several times to parameters such as A21, A22, etc., there is no information about their nature.
Response (5): Thank you for your reminder. We have explained and marked in red the symbols and abbreviations in the full text at their first occurrence.
Comment (6):
In the materials and methods section, it is better to show the schematic flow diagram for experiments.
Response (6): Thank you for your suggestion. We have added a schematic diagram of the flow for the experiment in Line 72 in the revised manuscript..
Comment (7):
It is better to arrange Figure 1 in a regular arrangement and in the form of one figure and one caption. In the current state, the titles are duplicated, and it has caused congestion. Also, Table 1. A and Table 1. B do not need to be separated.
Response (7): Thank you for your suggestion. We have converted all six graphs of Figure 1 to Table 2 and combined Table 1 A and Table 1B into a single Table 1.
Comment (8):
According to the results of parameters such as pH, moisture content and viscosity, is the beginning of detection of aerobic bacteria only related to lysozyme activity?
Response (8): Thank you. The main reason for detecting the total number of aerobic bacteria in the manscript is that most of the microorganisms that cause egg contamination belong to aerobic bacteria.
Comment (9):
Lines 212-227: It is better to discuss the reasons and mechanism of changes of freshness indicators during storage more (not necessarily in detail).
Response (9): Thank you. The reasons and mechanism of changes of freshness indicators during storage have been discussed in lines 230-235 in the revised manuscript.
.
Comment (10):
Table 3.A and 4.A: Please check the significance or non-significance of the correlation between freshness indicators and inserting one or two stars.
Response (10): Thank you. We have checked the significance or non-significance of the correlation between the freshness indicators and the insertion of one or two stars in Tables 3.A and 4.A and marked the modifications in red in the revised manuscript.
Comment (11):
What is Figure 5A in lines 308-309? Do you mean Table 5A?
Response (11): Thank you. Here should be Table 5A, which we have modified and marked red in the revised manuscript.
Comment (12):
Please introduce terms CSF and CSFM in Table 5.B.
Response (12): Thank you. May be due to two revisions made to the term "CFI" during the writing of the manuscript, CSF and CSFM in Table 5.B. should be CFI, that have been corrected in the revised manuscript.
Comment (13):
I think the gelling behavior of egg white during storage has not been properly discussed in terms of the effect of temperature and time. Apart from the correlation between freshness indicators and gelling properties, the mechanisms are not clear.
Response (13): Thank you so much. We further analyzed the reasons why the freshness of raw eggs varies with temperature and time lines 314-321; At the same time, we also strengthened the mechanism analysis of the correlation between CFI of the fresh duck eggs and gelling properties of the alkali-induced EWG gel lines 314-321.
Comment (14):
I could not find Figure 26 on page 15 (line 339).
Response (14): Thank you. It’s a miss that we accidentally wrote Table 6. incorrectly as Figure 26, which has been revised and marked in red in Line 327.
Comment (15):
The difference between abstract and conclusion is not respected and similar sentences are used. I recommend that according to the good results obtained in this research, the conclusion should be revised and the achievements reflected.
Response (15): Thank you so much. We have further improved the “Abstract and Conclusion” sections.
Reviewer 2 Report
Comments and Suggestions for Authors
Foods-2670069 Predict the quality of the preserved egg white based on the freshness of shell eggs
This manuscript provides important data for storage of duck shell eggs applicable in practice. Received equations can help to predict storage time for two temperatures 25 and 4 °C. After careful reading I can recommend minor changes only before acceptance for publishing. My comments are given in the following list.
- Lines 102-105 contain description of method of viscosity measurement. It has to be changed substantially. As I know egg albumen behaves as non-Newtonian fluid frequently thixotropic or structure destroying during measurement. For such a fluid there is necessary to provide geometry, temperature, shear rate at which the viscosity was measured valid.
- Line 122 parameters A21, A22 and A23 need to be explained here.
- Lines 137-139 are masked by the text “where r is the variance contribution”, please shift that to next line.
- Figure 1 E input gap at “albumen viscosity” and unit “mPa.s”. Change S to small letter s.
- Line 251 this equation is not numbered! Input equation number.
- Line 263 this equation is not numbered! Input equation number.
- Line 339 there is “Figure 26”. I can see only Figures 1 A up to F. Correct it.
- Line 344 input gap between “and” and “4°C”.
- Table 6 on page 16 contains heading “Storage at 25 °C” twice. I think that second part of the table is valid for 4 °C due to longer storage time.
- References have no doi codes, why?
Author Response
Dear editor and reviewers,
Thank you very much for your constructive and valuable suggestions. We have made careful modifications to the revised manuscript of the first edition. All changes made to the text are in red. Our responses to several comments are listed below:
Response to reviewer # 2
Comment (1):
Lines 102-105 contain description of method of viscosity measurement. It has to be changed substantially. As I know egg albumen behaves as non-Newtonian fluid frequently thixotropic or structure destroying during measurement. For such a fluid there is necessary to provide geometry, temperature, shear rate at which the viscosity was measured valid.
Response (1): Thank you very much for your suggestion. In the method of viscosity determination, we have added the geometry of the sample, temperature, and shear rate at the time of measurement.
Comment (2):
Line 122 parameters A21, A22 and A23 need to be explained here.
Response (2): Thank you. We have explained and marked in red the acronyms in the full text at their first occurrence in the revised manuscript.
Comment (3):
Lines 137-139 are masked by the text “where r is the variance contribution”, please shift that to next line.
Response (3): Thank you. We have adjusted the paragraph to the right place in the revised manuscript.
Comment (4):
Figure 1 E input gap at “albumen viscosity” and unit “mPa.s”. Change S to small letter s.
Response (4): Thank you very much. All these misses have been modified in the revised manuscript.
Comment (5):
Line 251 this equation is not numbered! Input equation number.
Response (5): Thank you. We have numbered this equation as “equation 4” in the revised manuscript.
Comment (6):
Line 263 this equation is not numbered! Input equation number.
Response (6): Thank you. We have numbered this equation as “equation 5” in the revised manuscript.
Comment (7):
Line 339 there is “Figure 26”. I can see only Figures 1 A up to F. Correct it.
Response (7): Thank you. We have corrected in the revised manuscript.
Comment (8):
Line 344 input gap between “and” and “4°C”.
Response (8): Thank you. We have input gap between “and” and “4°C” in the revised manuscript.
Comment (9):
Table 6 on page 16 contains heading “Storage at 25 °C” twice. I think that second part of the table is valid for 4 °C due to longer storage time.
Response (9): Thank you. We have corrected in the revised manuscript.
Comment (10):
References have no doi codes, why?
Response (10): Thank you. We have added doi codes for the reference in the revised manuscript.
Reviewer 3 Report
Comments and Suggestions for Authors
Interesting research and the data are partly novel. Some details in the methods should be elucidated.
83-86 Subsequently, a 20 g pre-treated albumen solution was mixed with 1 g NaOH and stirred at 200 r/min for a few minutes to let it stand to form gel. Finally, the alkali-induced EWG was placed at room temperature for 3 days to determine its gelling properties.
Albumen solution was mixed with solid NaOH? Local pH increase would decompose amino acids of albumen solution. What was the pH of obtained gels?
171-173 Alkali-induced EWG (10 mm in height and 20 mm in diameter) were compressed to 50% of their original height at pre-test speeds, test-speeds, and post-test speeds of 2.0 mm/s with a trigger point load of 5 g [19]. All three speeds were the same? Usually 1 mm/s is used for 10 mm samples.
The title should be changed. It is too general. Maybe: Correlation between the freshness of eggs and properties of alkali-induced egg white gel
English language should be polished by a native speaker. Some fragments are not fitting to the style.
99-101 Style…text pasted from a method description.
Comments on the Quality of English LanguageEnglish language should be polished by a native speaker. Some fragments are not fitting to the style.
99-101 Style…text pasted from a method description.
Author Response
Dear editor and reviewers,
Thank you very much for your constructive and valuable suggestions. We have made careful modifications to the revised manuscript of the first edition. All changes made to the text are in red. Our responses to several comments are listed below:
Response to reviewer # 3
Comment (1):
Albumen solution was mixed with solid NaOH? Local pH increase would decompose amino acids of albumen solution. What was the pH of obtained gels?
Response (1): Thank you. The addition of NaOH in the study was done in the form of a solution., we have stated the actual concentration of NaOH solution added to the egg whites in the revised manuscript. Moreover, the pH of the gel formed when the NaOH solution was added was 11.74 ± 0.06.
Comment (2):
171-173 Alkali-induced EWG (10 mm in height and 20 mm in diameter) were compressed to 50% of their original height at pre-test speeds, test-speeds, and post-test speeds of 2.0 mm/s with a trigger point load of 5 g [19]. All three speeds were the same? Usually 1 mm/s is used for 10 mm samples.
Response (2): Thank you. The parameter values for the measurement are established after many adjustments which are more suitable for our samples.
Comment (3):
The title should be changed. It is too general. Maybe: Correlation between the freshness of eggs and properties of alkali-induced egg white gel.
Response (3): Thank you, “Predict the gelling properties of alkali-induced egg white gel based on the freshness of duck eggs” was used to replace the “Predict the quality of the preserved egg white based on the freshness of shell eggs” in the revised manuscript.
Comment (4):
English language should be polished by a native speaker. Some fragments are not fitting to the style.
Response (4): Thank you very much. We have asked a native speaker to check the structure of manuscript and English language usage. We hope that the language is now meet the requirements of the journal.
Comment (5):
99-101 Style…text pasted from a method description.
Response (5): Thank you very much. We have rewritten this test method in the revised manuscript.
The manuscript has been resubmitted to your journal. We look forward to your attention.
Sincere Regards